# Mixed Integer Linear Programming Models to Solve a Real-Life Vehicle Routing Problem with Pickup and Delivery

Ali Louati [1,2,*], Rahma Lahyani [3], Abdulaziz Aldaej [1], Racem Mellouli [4,5] and Muneer Nusir [1]

1   Department of Information Systems, College of Computer Engineering and Sciences, Prince Sattam bin Abdulaziz University, Al-Kharj 11942, Saudi Arabia; a.adaej@psau.edu.sa (A.A.); m.nusir@psau.edu.sa (M.N.)
2   SMART Lab, ISG, University of Tunis, Tunis 2000, Tunisia
3   Operations and Project Management Department, College of Business, Alfaisal University, Riyadh 11533, Saudi Arabia; rlahyani@alfaisal.edu
4   Industrial Engineering Department, College of Engineering, Umm Al-Qura University, Al-Qunfudhah 24381, Saudi Arabia; ramellouli@uqu.edu.sa
5   Higher School of Business, University of Sfax, Sfax 3018, Tunisia
*   Correspondence: a.louati@psau.edu.sa

**Abstract:** This paper presents multiple readings to solve a vehicle routing problem with pickup and delivery (VRPPD) based on a real-life case study. Compared to theoretical problems, real-life ones are more difficult to address due to their richness and complexity. To handle multiple points of view in modeling our problem, we developed three different Mixed Integer Linear Programming (MILP) models, where each model covers particular constraints. The suggested models are designed for a mega poultry company in Tunisia, called CHAHIA. Our mission was to develop a prototype for CHAHIA that helps decision-makers find the best path for simultaneously delivering the company's products and collecting the empty boxes. Based on data provided by CHAHIA, we conducted computational experiments, which have shown interesting and promising results.

**Keywords:** vehicle routing problem; pickup and delivery; optimization; mixed integer linear programming

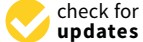



## 1. Introduction

Transportation studies can be considered as an intersection between several domains, methods, and techniques to propose, solve, and develop solutions for real-life problems. Artificial intelligence [1–4] and optimization [5–7] stand behind the majority of developed solutions. Among transportation and mobility problems, we recall the Vehicle Routing Problem (VRP). VRP is a well-known problem studied in Operational Research and Computational and Decision sciences. It consists of routing products from depots to customers by drawing adequate distribution plans. The routing plans, also called circuits or tours, need to respect multiple constraints, such as vehicle loading capacities and customer demands. The main objective of VRP is to provide significant savings in terms of transportation costs while ensuring the lowest delivery delays. A variant of VRP is called VRP with Pickup and Delivery (VRPPD). VRPPD joins the processes of collecting and delivering either simultaneously or separately. To deal with a real-life VRPPD, several requirements need to be considered, such as time window restriction, travel cost, vehicle capacity, heterogeneous vehicle fleet, vehicle travel time, vehicle speed, multi-dimensional capacity, route restrictions, and uncertain decisions to choose between picking up first and delivering second, or the opposite, or even performing both simultaneously. In addition, there might be different objective functions that could be minimized, such as those related to travel cost, satisfying customer requirements, loss of funds, etc.

This study is motivated by an industrial context, which is a problem related to a mega poultry company in Tunisia called CHAHIA. CHAHIA supplies a large number of

meat product outlets every day. The various orders issued daily to serve these outlets are delivered in boxes owned by CHAHIA. The logistics of product delivery and box pickup are complicated and generates high costs since boxes are subject to loss (according to CHAHIA). Such losses could lead to stock shortages when filling and preparing products for delivery. Therefore, these boxes require a full-fledged management process when modeling the VRPPD for the company. The real VRPPD that we are dealing with differs from traditional VRPPDs in several aspects. First, our proposed models allow multiple visits to customers on the same day. In addition, each customer could be served by multiple vehicles during the same day. Furthermore, we have included CHAHIA's specificities, which are not well addressed in the literature.

This paper develops three different Mixed Integer Linear Programming models (MILP) for CHAHIA VRPPD, handling multiple points of view. The first model is a VRP with a Simultaneous Pickup and Delivery (VRPSPD). It ensures that the pickup and delivery are performed simultaneously, without leaving any empty boxes with any customers. The second model is a flexible VRPPD. It provides two options for pickup and delivery operations, separately and/or simultaneously. The third one is a VRPSPD ensuring simultaneous pickup and delivery. However, it provides flexibility to drop and leave some boxes, which will be consequently penalized. The proposed models are benchmarked based on data provided by CHAHIA. We claim that in practice, defining a VRPPD problem is much more laborious than solving it due to the fact that the same problem could be viewed from different angles, observations, perspectives, and preferences. The multiple VRPPD readings could lead to different objective functions by focusing or ignoring certain details, data aggregations, and setting assumptions (see Figure 1).

The remainder of this paper is organized as follows. Section 2 reviews the related literature. Section 3 describes the VRPSPD (*MILP*1). The flexible VRPPD (*MILP*2) is detailed in Section 4. Section 5 describes the VRPSPD with two additional alternatives which are, ignoring pickups and penalizing the loss (*MILP*3). Section 5 details the experimental framework. Section 6 discusses the limitation of the paper. Finally, Section 7 concludes the paper and opens new perspectives.

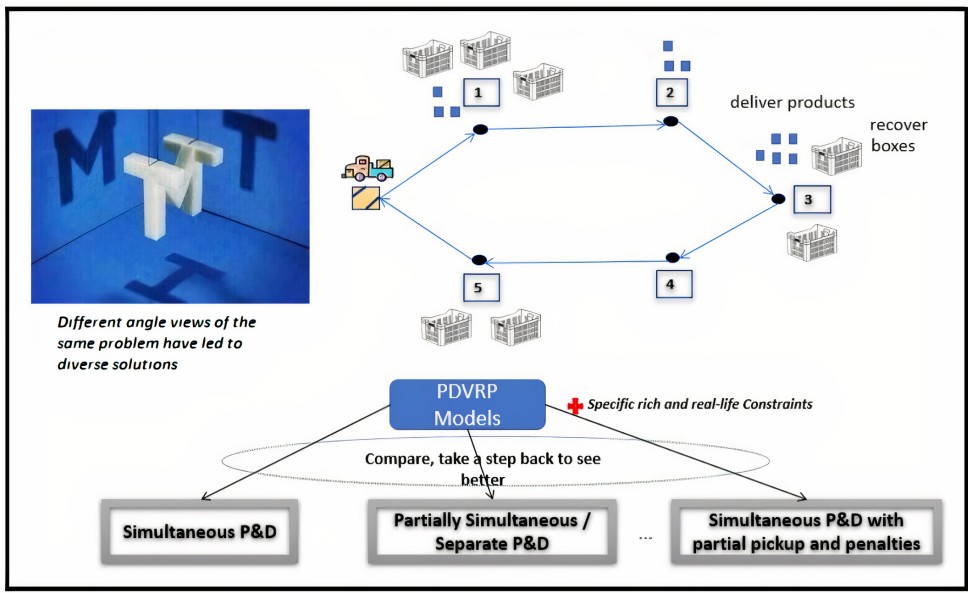

**Figure 1.** Multiple readings of the same problem.

## 2. Related Literature

The VRPPD could be classified into three different categories [8,9], which are as follows:

- **Delivery first and pickup second**: vehicles perform a pickup operation after the delivery process. This category is called Vehicle Routing Problems with Backhauls (VRPB).
- **Mixed pickup and delivery**: vehicles deliver or pickup in any sequence along their routes. This category is called Mixed Vehicle Routing Problem with Backhauls (MVRPB).
- **Simultaneous Pickup and delivery**: vehicles simultaneously perform the delivery and pickup. This category is called Vehicle Routing Problems with Simultaneous Pickup and Delivery (VRPSPD).

The majority of published VRPPD models assume that each customer is visited only once [10]. Some exceptions have considered multi-customer visits, such as in [11] where the authors developed a large-scale model, allowing two visits per customer for a many-to-many pickup and delivery routing problem. In [12,13], the authors considered a VRPSPD involving the delivery of full bottles and the collection of empty ones. The authors assumed that such a problem is classified among reverse logistics operations.

Several real-life applications encountered in the beverage industry are described in [14]. For example, in [15], authors considered a truck scheduling problem for container transportation in a local area with multiple depots and terminals. They proposed an approach based on an integer programming heuristic that determines pickup and delivery sequences for daily drayage operations while ensuring a minimum transportation cost. In the same context, the authors in [16] demonstrated that drayage operations could be considered as a multi-stop VRPB. More details regarding VRPB and VRPPD can be found in two comprehensive overviews provided in [17,18], respectively. The problem of container drayage has been considered by other research work. For example, in [19], authors have developed models capable of minimizing both present and future operating costs. The authors in [20] modeled the container drayage problem as a pickup and delivery problem and proposed Lagrangian relaxation to solve the problem. In [21], the authors proposed three approaches for the container movement problem with time windows at origins and destinations. These approaches are based on an asymmetric multiple traveling salesman problem.

Some researchers have considered a single vehicle while modeling the VRPPD. For example, the single-vehicle VRPPD with deterministic demands and predetermined customer visit sequence has been considered in [22]. Authors in [22] have developed the routing of a single-vehicle that delivers multiple products under stochastic demand. Other researchers, such as [23,24], have considered a VRP with mixed pickup and delivery, called (VRPM), where pickups are made before deliveries.

Recently, Zhang et al. [25] introduced a new VRP variant which encourages the reuse of collected items. They developed a segment-based evaluation procedure to reduce the computation time. They provided a mathematical formulation and a metaheuristic algorithm, and analyzed several features of the problem. In [26], a hybrid heuristic algorithm for the 3L-PDP problem is extended by using two key improvements: the first is the usage of a tabu strategy for enlarging the local search space, the second is the employment of complex block generation and depth-first heuristics for incrementally finding one proper box at a time in the packing phase. The experimental results show that the improved hybrid heuristic algorithm outperforms its origin regarding total travel distance on benchmark state-of-the-art instances. More variants of VRPPD including MRVRP, VRPP, SDVRP, SVRP, VRPSF, and VRPTZ can be found detailed in [27,28].

Finally, we recall the main contributions of this paper. We have been motivated by the large number of studies that tackled the VRPPD. It is worth noting that the majority of possible constraints have already been suggested in the last two decades. However, our work provides prototypes to solve the VRPPD based on CHAHIA's needs. These needs include three different variants of VRPPD. The first variant is a VRP with a Simultaneous Pickup and Delivery (VRPSPD) without leaving any empty boxes at any customer. The second variant is a flexible VRPPD that ensures separate and/or simultaneous pickup and

delivery. The third one is a simultaneous pickup and delivery VRP allowing the flexibility to drop and leave some boxes, which will be consequently penalized. More specificities regarding the proposed models can be found in Section 3.

## 3. Problem Ingredients and Approach of the Proposed MILP Models

In order to state the problem more formally and clearly, we present our ideas which are based on a real-life context. We have taken into account a set of assumptions, which are as follows:

- **Time window**:
  Includes the service time at each customer, loading time for each vehicle, travel time between each couple of customers, time necessary to park the vehicle, and the workday duration.
- **Combined capacity; volume and weight**:
  The majority of VRP formulations consider the physical capacity of vehicles, which is generally expressed in terms of weight. In our problem, goods are delivered using boxes. Therefore, beside the capacity in terms of weight, we have to make sure that each vehicle has the capacity in terms of number of boxes to be fitted into the vehicle. Thus, we defined a new volume capacity that includes both loaded weight and box numbers. We denote that we have considered the volume occupation in each vehicle to be discrete. Motivated by this assumption, which complicates the use of vehicle space, we assume that the combined capacity has not been addressed in the literature.
- **Customer balance (stock) in terms of boxes**:
  We suppose, initially, that each customer has boxes ready for pickup. These boxes are considered by our models before performing any delivery.
- **Optimization choice**:
  Three different assumptions are considered. The first one guarantees that the pickup and delivery are performed simultaneously, without leaving any empty boxes for any customer. The second assumption is more flexible. It provides two choices of pickup and delivery operation, separately and/or simultaneously. The third one ensures that the pickup and delivery operation are performed simultaneously. However, it provides a flexibility to drop and leave some boxes, which will be consequently penalized.
- **Heterogeneity of the fleet**
  Multiple vehicle types exist, having different sizes.

## 4. Mathematical Formulations

Three different formulations of the real-life VRPPD are detailed in this section.

### 4.1. Simultaneous Pickup and Delivery Vehicle Routing Problem: MILP 1

This section models the Simultaneous Pickup and Delivery Vehicle Routing Problem variant (SPDVRP). In SPDVRP, the operations of pickup and delivery are performed simultaneously at each customer node. Each customer is visited once, their order (in terms of goods) is all delivered, and all boxes are picked up. The variables and parameters of the proposed model are as follows:

**- Indices and parameters**:

- $n$: number of customers ($i = 1, \ldots, n$).
- $m$: number of vehicles ($i = 1, \ldots, m$).
- $d_i^{(w)}$: customer $i$ demand in $Kg$.
- $d_i^{(c)}$: volume demand of customer $i$ in terms of number of boxes. This parameter is deduced from the following relation: $d_i^{(c)} = \frac{d_i^{(w)}}{\alpha_i}$.
- $\alpha_i$: average weight of a full loaded box dedicated for a customer $i$.
- $Sc_i^{ini}$: initial balance of boxes that exists at the store of a customer $i$.

- $\beta$: weight of an empty box.
- $t_{ij}$: travel time between two customers $i$ and $j$.
- $TS_u$: unitary time service expressed in minute per box.
- $TS_f$: fixed service time.
- $TS_j$: total service time at a customer $i$.
- $TC_j$: time necessary to load each vehicle at the depot.
- $TPC_u$: unitary time needed to prepare the boxes expressed in terms of minute per box.
- $T_{max}$: maximum duration of a tour.
- $Q_k^{(w)}$: weight capacity of a vehicle $k$ expressed in *number of boxes*.
- $Q_k^{(c)}$: volume capacity of a vehicle $k$ expressed in *Kg*.
- $C_{ij}$: travel cost from a customer $i$ to another $j$.

**- Decision variables**:

- $W_j$: weight of the vehicle load before leaving a customer $j$.
- $W_{(0,k)}$: weight of goods at the depot for each vehicle $k$.
- $C_j$: number of boxes before leaving a customer $j$.
- $C_{(0,k)}$: number of boxes loaded at the depot for a vehicle $k$.
- $U_j$: variable used for sub-tours elimination.
- $x_{ij}^k = \begin{cases} 1 \text{ if the vehicle travels from } i \text{ to } j. \\ i,j = 1,...,n(i \neq j); \forall k = 1,...,m \\ 0 \text{ otherwise} \end{cases}$

The $MILP1$ formulation is as follows:

$$Min \sum_{k=1}^{m} \sum_{j=0}^{n} \sum_{i=0_{i \neq j}}^{n} c_{ij} x_{ij}^k \quad (1a)$$

$$\sum_{k=1}^{m} \sum_{i=0_{i \neq j}}^{n} x_{ij}^k = 1 \quad (\forall j = 1,...,n) \quad (1b)$$

$$\sum_{i=0_{i \neq j}}^{n} x_{ji}^k = \sum_{i=0_{i \neq j}}^{n} x_{ij}^k \quad (\forall j = 1,...,n; \forall k = 1,...,m) \quad (1c)$$

$$\sum_{j=0_{i \neq j}}^{n} \sum_{i=0_{i \neq j}}^{n} (t_{ij} + TS_j + TC_j) x_{ij}^k \leqslant T_{(max)} \quad (\forall k = 1,...,m) \quad (1d)$$

$$W_j \leqslant \sum_{k=1}^{m} \left[ Q_K^{(W)} \sum_{i=0_{i \neq j}}^{n} x_{ij}^k \right] (\forall j = 1,...,n) \quad (1e)$$

$$W_{(0,k)} \leq Q_k^{(w)} (\forall k = 1,...,m) \quad (1f)$$

$$C_j \leq \sum_{k=1}^{m} \left[ Q_k^{(c)} \sum_{i=0_{i \neq j}}^{n} x_{ji}^k \right] (\forall j = 1,...,n) \quad (1g)$$

$$C_{(0,k)} \leq Q_k^{(c)} (\forall k = 1,...,m) \quad (1h)$$

$$W_j - W_i \geq -d_j^{(w)} + (Sc_j^{ini} + d_j^{(c)}) * \beta + (\sum_{k=1}^{m} x_{ij}^k - 1) M_9 \quad (\forall i,j = 1,...,n, i \neq j) \quad (1i)$$

$$W_j - W_i \geq -d_j^{(w)} + (Sc_j^{ini} + d_j^{(c)}) * \beta + (1 - \sum_{k=1}^{m} x_{ij}^k) M_{10} (\forall i,j = 1,...,n, i \neq j) \quad (1j)$$

$$W_{(0,k)} = \sum_{j=1}^{n} d_j^{(w)} * \sum_{i=0_{i \neq j}}^{n} x_{ij}^k (\forall k = 1,...,m) \quad (1k)$$

$$W_j - W_{(0,k)} \geq -d_j^{(w)} + (Sc_j^{ini} + d_j^{(c)}) * \beta + (x_{0j}^k - 1) M_{12} (\forall j = 1,...,n; \forall k = 1,...,m) \quad (1l)$$

$$W_j - W_{(0,k)} \geq -d_j^{(w)} + (Sc_j^{ini} + d_j^{(c)}) * \beta + (x_{0j}^k - 1) M_{13} (\forall j = 1,...,n; \forall k = 1,...,m) \quad (1m)$$

$$C_j - C_i \leq Sc_j^{ini} + (1 - \sum_{k=1}^{m} x_{ij}^k)M14 (\forall i,j = 1,...,n, i \neq j) \qquad (1\text{n})$$

$$C_j - C_i \leq Sc_j^{ini} + (1 - \sum_{k=1}^{m} x_{ij}^k)M15 (\forall i,j = 1,...,n, i \neq j) \qquad (1\text{o})$$

$$C_{(0,k)} = \sum_{j=1}^{n} d_j^{(c)} * \sum_{i=0}^{n} x_{ij_{i \neq j}}^k \quad (\forall k = 1,...,m) \qquad (1\text{p})$$

$$C_j - C_{(0,k)} \geqslant Sc_j^{ini} + (x_{0j}^k - 1)M_{17} (\forall j = 1,...,n; \ \forall k = 1,...,m) \qquad (1\text{q})$$

$$C_j - C_{(0,k)} \geqslant Sc_j^{ini} + (1 - x_{0j}^k)M_{18} (\forall j = 1,...,n; \ \forall k = 1,...,m) \qquad (1\text{r})$$

$$U_j - U_i + \sum_{k=1}^{m} \left[ M_{19} + d_i^{(w)} x_{ij}^k + (-d_j^{(w)} + M_{19}) x_{ij}^k \right] \leq M_{19} \ (\forall i,j = 1,...,n; \ i \neq j) \qquad (1\text{s})$$

$$d_j^{(w)} \leq U_j \leq \sum_{k=1}^{m} \sum_{i=0_{i \neq j}}^{n} Q_k^{(w)} x_{ij}^k \qquad (1\text{t})$$

Knowing that:

- $TS_j = TS_{(u)}(2d_j^{(c)} + Sc_j^{ini}) + TS_{(f)}$
- $TC_j = TPc_{(u)} * d_j^{(c)}$
- $M_9 = M_{10} = M_{12} = M_{13} = M_{19} = Max_{k'} Q_{k'}^{(w)}$
- $M_{14} = M_{15} = M_{17} = M_{18} = Max_{k'} Q_{k'}^{(c)}$

According to the *MILP*1 formulation, the objective function (1a) seeks to minimize the total transportation cost. Constraints (1b) and (1c) ensure that the first vertex (node) associated with each customer is visited once for a simultaneous pickup and delivery. Constraint (1d) is related to the time service. The latter includes the travel time between two customers and the time necessary for loading goods into the vehicle. It prevent exceeding the predefined time capacity. Constraint (1e) ensures that the weight of the vehicle load before leaving a customer $j$ does not exceed the vehicle capacity if there is a path between customers $i$ and $j$. Constraint (1f) ensures that the weight of goods required by all customers does not exceed the vehicle capacity in terms of weight. Constraint (1g) guarantees that the number of boxes loaded into the vehicle before leaving a customer $j$ does not exceed the vehicle volume whenever it travels from $j$ to another customer $i$. Constraint (1h) ensures that the number of boxes required by all customers does not exceed the vehicle volume capacity. Constraints (1i) and (1j) express the relation between two successive customers in terms of weight. We considered here two types of weight; one is associated with the delivery boxes, while the other one is associated with the initial boxes existing at each customer store. Constraint (1k) defines the vehicle load before leaving the depot. Constraints (1l) and (1m) present the weight of goods loaded into the vehicle before traveling from the depot to a customer $j$. Constraints (1n) and (1o) represent the relation between two successive customers in terms of volume (number of boxes). Constraint (1p) defines the boxes loaded into the vehicle before leaving the depot. Constraints (1q) and (1r) show the relation between the depot and a customer $j$ in terms of volume. Finally, (1s) and (1t) represent the sub-tour elimination constraints.

### 4.2. Flexible VRPPD: MILP 2

*MILP*2 is more flexible than *MILP*1. It considers a different representation of customer services. We associate two vertices $i$ and $i + n$ with each customer $i$, where $i$ and $i + n$ are the same, but have different types of service operations.

In reality, customers can be classified into two types according to the way of required services. Indeed, *MILP*2 allows two types of services for customer $i$. The first type assumes that the pickup and delivery are performed simultaneously. In such cases, vertex $i$ is visited and vertex $i + n$ is left. The second type ensures that a customer $i$ is visited twice, where the

delivery is performed at vertex $i$, while pickup is done at vertex $i + n$. Figure 2 illustrates the representation of customers based on a virtual replica.

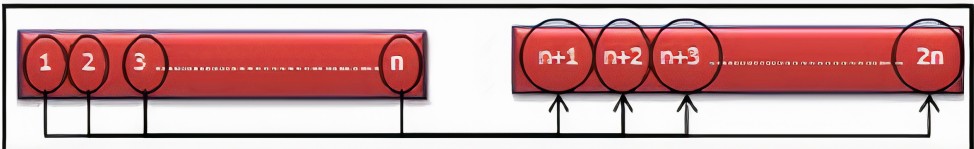

**Figure 2.** Representation of the virtual customers $(n + 1, ..., 2n)$.

Figure 3 represents two feasible tours for the same vehicle trip. The solution includes nodes visited once with simultaneous pickup and delivery (black nodes), and nodes visited twice due to two separated operations (encircled black nodes). The originality in the proposed configuration is that certain feasible subtours must be allowed in appearance. However, thanks to the duplication of nodes, it became easy to model the problem by conserving classical subtour elimination principles, while using a subtle manner of formulation that takes into account the non-conservative progression of vehicle load.

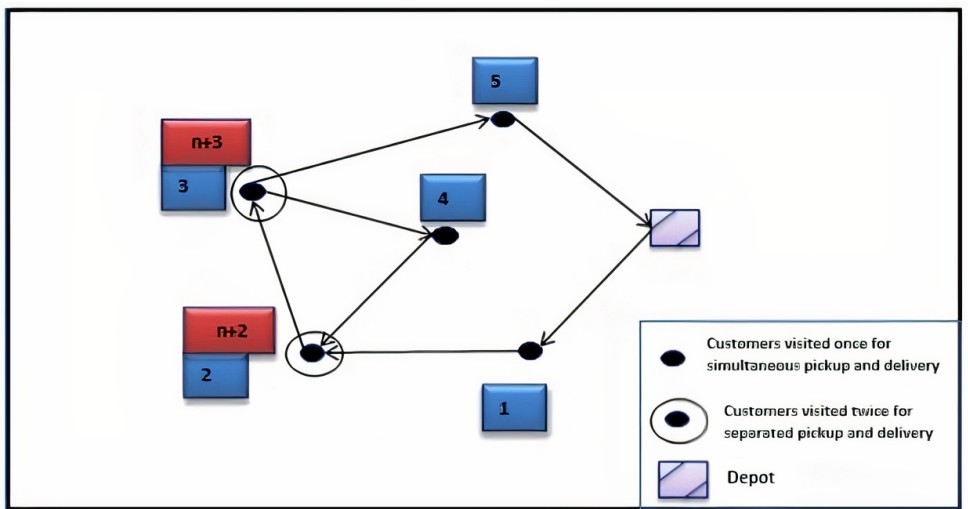

**Figure 3.** Overview of the flexible vehicle routing problem with pickup and delivery (VRPPD): *MILP*2.

For *MILP*2 formulation, we define additional parameters and variables, which are as follows.

**Additional parameters for *MILP*2:**

- $T_{(0,k)}$: vehicle availability time at the depot. $\forall k = 1, .. m$.
- $T_j$: flow time at customer $j$.

**Additional variables for *MILP*2:**

- $y_i = \begin{cases} 1\ if\ pickup\ and\ delivery\ are\ performed\ simultaneously\ at\ customer\ j.(j = 1, ..., n) \\ 0\ otherwise, \end{cases}$

- $y_i = \begin{cases} 1\ if\ the\ vehicle\ travels\ from\ i\ to\ j \\ (j = n + 1, ..., 2n;\ \forall i = 1, .. 2n;\ i \neq j;\ \forall k = 1, .. m) \\ 0\ otherwise, \end{cases}$

*MILP*2 formulation is as follows:

$$Min \sum_{k=1}^{m} \sum_{j=0}^{n} \sum_{i=0_{i \neq j}}^{n} c_{ij} x_{ij}^{k} \qquad (2a)$$

$$\sum_{k=1}^{m}\sum_{i=0_{i\neq j}}^{n} x_{ij}^{k} = 1 \quad (\forall j = 1, ..., n) \qquad (2b)$$

$$\sum_{i=0}^{2n} x_{ji}^{k} = \sum_{i=0}^{2n} x_{ij}^{k} \ \forall j = 1, ..., 2n; \ \forall k = 1, ..., m \qquad (2c)$$

$$\sum_{k=1}^{m}\sum_{i=0_{i\neq j}}^{2n} x_{ij}^{k} = 1 - y_{j-n} \quad \forall j = n+1, ..., 2n \qquad (2d)$$

$$\sum_{j=n+1}^{2n}\left[\sum_{j=0_{j\neq i}}^{0}(t_{ij}+TS_{(f)})x_{ij}^{k}+\sum_{j=1_{j\neq i}}^{n}(TS_{(u)}*d_{j}^{(c)}+y_{j}(d_{j}^{(c)}+Sc_{j}^{ini})*TS_{(u)})x_{ij}^{k}+\right.$$

$$\left.\sum_{j=n+1}^{2n}(t_{ij}+1(1-y_{j-1})*\left[TS_{(f)}+TS_{TS(u)}(d_{j}^{(c)}+Sc_{j}^{ini})\right]+TC_{j})x_{ij}^{k}\right]\le T_{((max))}$$

$$\forall(k = 1, ..., m) \qquad (2e)$$

$$W_{j} \le \sum_{k=1}^{m}\left[Q_{k}^{(w)}\sum_{i=0_{i\neq j}}^{2n} x_{ij}^{k}\right] \quad (\forall j = 1, ..., 2n) \qquad (2f)$$

$$W_{(0,k)} \le Q_{k}^{(w)} \quad (\forall k = 1, ..., m) \qquad (2g)$$

$$C_{j} \le \sum_{k=1}^{m}\left[Q_{k}^{(c)}\sum_{i=0_{i\neq j}}^{2n} x_{ij}^{k}\right] \quad (\forall j = 1, ..., 2n) \qquad (2h)$$

$$C_{(0,k)} \le Q_{k}^{(c)} \quad (\forall k = 1, ..., m) \qquad (2i)$$

$$W_{j} - W_{i} \ge -d_{j}^{(w)} + (Sc_{j}^{ini}+d_{j}^{(c)})*\beta*y_{i}+(\sum_{k=1}^{m} x_{ij}^{k}-1)M_{10}$$

$$(\forall j = 1, ..., 2n; \ \forall i = 1, ..., n; \ i \neq j) \qquad (2j)$$

$$W_{j} - W_{i} \ge -d_{j}^{(w)} + (Sc_{j}^{ini}+d_{j}^{(c)})*\beta*y_{i}+(1-\sum_{k=1}^{m} x_{ij}^{k})M_{11}$$

$$(\forall j = 1, ..., 2n; \ \forall i = 1, ..., n; \ i \neq j) \qquad (2k)$$

$$W_{j} - W_{i} \ge (Sc_{j-n}^{ini}+d_{j-n}^{(c)})*\beta+(1-\sum_{k=1}^{m} x_{ij}^{k}+y_{j-n})M_{12}$$

$$(\forall j = n+1, ..., 2n; \ \forall i = 1, ..., 2n; \ i \neq j) \qquad (2l)$$

$$W_{j} - W_{i} \ge (Sc_{j-n}^{ini}+d_{j-n}^{(c)})*\beta+(\sum_{k=1}^{m} x_{ij}^{k}+y_{j-n}-1)M_{13}$$

$$(\forall j = n+1, ..., 2n; \ \forall i = 1, ..., 2n; \ i \neq j) \qquad (2m)$$

$$W_{(0,k)} = \sum_{j=1}^{n}d_{j}^{(w)}*\sum_{i=0_{i\neq j}}^{n} x_{ij}^{k} \quad (\forall k = 1, ..., m) \qquad (2n)$$

$$W_{j} - W_{(0,k)} \ge -d_{j}^{(w)} + (Sc_{j}^{ini}+d_{j}^{(c)})*\beta*y_{i}+(x_{0j}^{k}-1)M_{15}$$

$$(\forall j = 1, ..., n; \ \forall k = 1, ..., m) \qquad (2o)$$

$$W_{j} - W_{(0,k)} \ge -d_{j}^{(w)} + (Sc_{j}^{ini}+d_{j}^{(c)})*\beta*y_{i}+(1-x_{0j}^{k})M_{16}$$

$$(\forall j = 1, ..., n; \ \forall k = 1, ..., m) \qquad (2p)$$

$$W_{j} - W_{(0,k)} \ge (Sc_{j-n}^{ini}+d_{j-n}^{(c)})*\beta+(1-x_{0j}^{k}+y_{j-n})M_{17}$$

$$(\forall j = n+1, ..., 2n; \ \forall k = 1, ..., m) \qquad (2q)$$

$$W_j - W_{(0,k)} \geq (Sc_{j-n}^{ini} + d_{j-n}^{(c)}) * \beta + (x_{0j}^k - 1 - y_{j-n})M_{18}$$
$$(\forall j = n+1, ..., 2n; \ \forall k = 1, ..., m) \qquad (2r)$$

$$C_j - C_i \leq Sc_j^{ini} * y_i + (\sum_{k=1}^m x_{ij}^k - 1)M19 \quad (\forall j = 1, ..., n; \ \forall i = 1, ..., 2n; \ i \neq j) \qquad (2s)$$

$$C_j - C_i \leq Sc_{j-n}^{ini} * y_j + (1 - \sum_{k=1}^m x_{ij}^k)M20 \quad (\forall j = 1, ..., n; \ \forall i = 1, ..., 2n; \ i \neq j) \qquad (2t)$$

$$C_j - C_i \leq Sc_{j-n}^{ini} + (1 - \sum_{k=1}^m x_{ij}^k + y_{j-n})M21 \quad (\forall j = 1, ..., n; \ \forall i = 1, ..., 2n; \ i \neq j) \qquad (2u)$$

$$C_j - C_i \geq Sc_{j-n}^{ini} + (\sum_{k=1}^m x_{ij}^k - 1 - y_{j-n})M22$$
$$(\forall j = n+1, ..., 2n; \ \forall i = 1, ..., 2n; \ i \neq j; \ k = 1, ..., m \qquad (2v)$$

$$C_{(0,k)} = \sum_{j=1}^n d_j^{(c)} * \sum_{i=0_{i \neq j}}^n x_{ij}^k \quad (\forall k = 1, ..., m \ ) \qquad (2w)$$

$$C_j - C_{(0,k)} \geq Sc_j^{ini} * y_j + (x_{0j}^k - 1)M24$$
$$(\forall j = 1, ..., n; \ \forall i = 1, ..., n; \ \forall k = 1, ..., m; i \neq j) \qquad (2x)$$

$$C_j - C_{(0,k)} \leq Sc_j^{ini} * y_j + (1 - x_{0j}^k)M25$$
$$(\forall j = 1, ..., n; \ \forall i = 1, ..., n; \ \forall k = 1, ..., m; i \neq j) \qquad (2y)$$

$$C_j - C_{(0,k)} \geq Sc_{j-n}^{ini}(x_{0j}^k - 1)M26$$
$$(\forall j = 1, ..., n; \ \forall j = n+1, ..., 2n; \ \forall k = 1, ..., m) \qquad (2z)$$

$$C_j - C_{(0,k)} \leq Sc_{j-n}^{ini}(1 - x_{0j}^k)M27$$
$$(\forall j = 1, ..., n; \ \forall j = n+1, ..., 2n; \ \forall k = 1, ..., m) \qquad (2aa)$$

$$T_j - T_i \leq TS_i + \sum_{k=1}^m t_{ij}x_{ij}^k + (1 - \sum_{k=1}^m x_{ij}^k)M_{28}(\forall j = 1, ..., n; \ \forall i, j = 1, ..., 2n; \ i \neq j) \qquad (2ab)$$

$$T_j - T_i \geq TS_i + \sum_{k=1}^m t_{ij}x_{ij}^k + (\sum_{k=1}^m x_{ij}^k - 1)M_{29}(\forall j = 1, ..., n; \ \forall i, j = 1, ..., 2n; \ i \neq j) \qquad (2ac)$$

$$T_j \leq \left[ \sum_{k=1}^m \sum_{i=0}^n \sum_{i'=0}^{2n} x_{i'i}^k TC_i + TS_0 + \sum_{K=1}^M (t_{0j} + T_{(0,k)}x_{0j}^k) \right] + (1 - \sum_{k=1}^m x_{0j}^k)M_{30}$$
$$(\forall j = 1, ..., 2n) \qquad (2ad)$$

$$T_j \geq \left[ \sum_{k=1}^m \sum_{i=0}^n \sum_{i'=0}^{2n} x_{i'i}^k TC_i + TS_0 + \sum_{K=1}^M (t_{0j} + T_{(0,k)}x_{0j}^k) \right] + (\sum_{k=1}^m x_{0j}^k - 1)M_{31}$$
$$(\forall j = 1, ..., 2n) \qquad (2ae)$$

$$T_j \leq T_{j+n} + y_j T_{max}(\forall j = 1, ..., n) \qquad (2af)$$

$$T_{j+n} \leq (1 - y_j)T_{max}(\forall j = 1, ..., n) \qquad (2ag)$$

$$U_j - U_i + \sum_{k=1}^m (M_{34} + 1)x_{ij}^k + (M_{34} - 1)x_{ij}^k \leq M_{34} \quad (\forall i, j = 1, ..., 2n; \ i \neq j) \qquad (2ah)$$

$$1 \leq U_j \leq 2n \forall i, j = 1, ..., 2n) \qquad (2ai)$$

Knowing that:

- $TC_j = TPc_u * d_j^{(c)}$
- $M_{10} = M_{11} = M_{12} = M_{13} = M_{14} = M_{15} = M_{16} = M_{17} = M_{18} = M_{28} = M_{29} = M_{30} = M_{31} = M_{34} = Max_{k'}Q_{k'}^{(w)}$
- $M_{19} = M_{20} = M_{21} = M_{22} = M_{24} = M_{25} = M_{26} = M_{27} = M_{34} = Max_{k'}Q_{k'}^{(c)}$

$MILP2$ shares the same objective function as $MILP1$. However, constraints (2b) and (2c) mean that the first vertex associated with each customer must be visited once, either for a single operation of delivery or for a simultaneous pickup and delivery. Constraint (2d) ensures that the second vertex associated with the customer is visited only if a combined pickup and delivery do not occur at the first vertex. Constraint (2e) defines the time capacity. It contains three parts which are; the time capacity at the depot, the time capacity when the pickup and delivery are performed simultaneously, and the time capacity when the pickup are performed separately. Both Constraints (2l) and (2m) define $Wj$ in terms of $Wi$ whenever $j$ is visited immediately after $i$, knowing that $i$ and $j$ are two virtual customers. Constraints (2q) and (2r) express the relation between a customer $j$ and the depot in terms of weight. Both constraints define $Wj$ in terms of $W(0, k)$ whenever $j$ is visited immediately after the depot, knowing that $j$ represents a virtual customer. Constraints (2z) and (2aa) express the link between a customer $j$ and the depot in terms of volume (number of boxes). Constraints (2ab) and (2ac) define $Tj$ in terms of $Ti$ whenever $j$ is visited immediately after $i$. Both constraints represent the time left between $i$ and $j$. Constraints (2ad) and (2ae) express the time relation between a customer $j$ and the depot. Constraints (2af) and (2ag) represent the time relation at a customer $j$. Both constraints reveal whether the pickup and delivery are performed simultaneously or not. Finally, Constraints (2ah) and (2ai) represent the subtour elimination constraints. We note that $yj$ is considered in $MILP1$ to make sure of whether the pickup and delivery are performed simultaneously or not.

### 4.3. Flexible VRPSPD: MILP 3

$MILP3$ represents a simultaneous pickup and delivery operation such as the one in $MILP1$. However, the novelty in $MILP3$ is its flexibility, which allows the dropping/releasing of some boxes. Thus, a partial collection of boxes is acceptable due to either a limited vehicle capacity and/or the high cost of collecting all boxes. Despite $MILP3$ being more realistic in terms of collecting flexibility, several trips become necessary to collect all boxes, which could be high in terms of cost. To deal with this issue, we added a penalty cost to the objective function that penalizes the process of leaving boxes. Figure 4 represents an overview of $MILP3$.

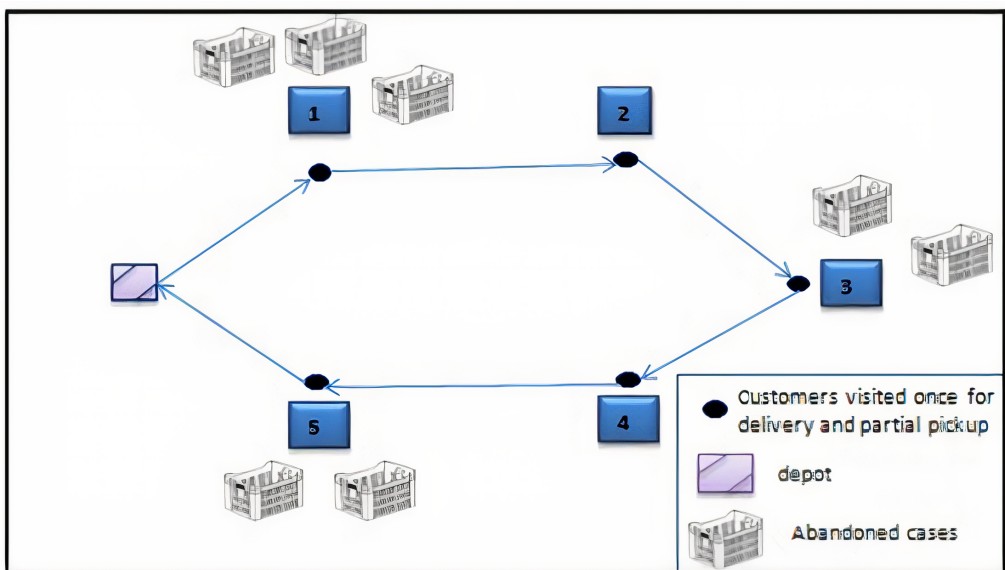

**Figure 4.** Overview of the flexible VRPSPD: $MILP3$.

We define the following additional variables for $MILP3$.

- $Sc_j^{end}$: boxes left at customer $j$.
- $c_{ca}$: penalizing cost associated to the left boxes.

*MILP*3 formulation is as follows:

$$Min \sum_{k=1}^{m} \sum_{j=0}^{n} \sum_{i \neq j}^{n} c_{ij} x_{ij}^{k} + \sum_{j=1}^{n} c_{ca} Sc_{j}^{end} \qquad (3a)$$

$$\sum_{k=1}^{m} \sum_{i=0_{i \neq j}}^{n} x_{ij}^{k} = 1 \quad (\forall j = 1, \dots n) \qquad (3b)$$

$$\sum_{i=0_{i \neq j}}^{n} x_{ji}^{k} = \sum_{i=0_{i \neq j}}^{n} x_{ij}^{k} \quad (\forall j = 1, \dots n; \ \forall k = 1, \dots m) \qquad (3c)$$

$$\sum_{i=0_{i \neq j}}^{n} x_{ji}^{k} = \sum_{i=0_{i \neq j}}^{n} x_{ij}^{k} \quad (\forall j = 1, \dots n; \ \forall k = 1, \dots m) \qquad (3d)$$

$$\sum_{j=0}^{n} \sum_{i=0_{i \neq j}}^{n} (t_{ij} + TS_j + TC_j) x_{ij}^{k} \leq T_{max} \quad (\ \forall k = 1, \dots m) \qquad (3e)$$

$$W_j \leq \sum_{k=1}^{m} \left[ Q_k^{(w)} \sum_{i=0_{i \neq j}}^{n} x_{ij}^{k} \right] \quad (\forall j = 1, \dots, n) \qquad (3f)$$

$$W_{(0,k)} \leq Q_k^{(w)} \quad (\forall k = 1, \dots, m) \qquad (3g)$$

$$C_j \leq \sum_{k=1}^{m} \left[ Q_k^{(c)} \sum_{i=0_{i \neq j}}^{n} x_{ij}^{k} \right] \quad (\forall k = 1, \dots, n) \qquad (3h)$$

$$C_{(0,k)} \leq Q_k^{(C)} \quad (\forall j = 1, \dots, m) \qquad (3i)$$

$$W_j - W_i \geq -d_j^{(w)} + (Sc_j^{ini} + d_j^{(c)} - Sc_j^{end}) * \beta + (\sum_{k=1}^{m} x_{ij}^{k} - 1) M_9$$
$$(\forall i, j = 1, \dots, n; \ i \neq j) \qquad (3j)$$

$$W_j - W_i \geq -d_j^{(w)} + (Sc_j^{ini} + d_j^{(c)} - Sc_j^{end}) * \beta + (1 - \sum_{k=1}^{m} x_{ij}^{k}) M_{10}$$
$$(\forall i, j = 1, \dots, n; \ i \neq j) \qquad (3k)$$

$$W_{(0,k)} = \sum_{j=1}^{n} d_j^{(w)} \sum_{i=0_{i \neq j}}^{n} x_{ij}^{k} \quad (\forall k = 1, \dots, m) \qquad (3l)$$

$$W_j - W_{(0,k)} \geq -d_j^{(w)} + (Sc_j^{ini} + d_j^{(c)} - Sc_j^{end}) * \beta + (x_{0j}^{k} - 1) M_{12}$$
$$(\forall j = 1, \dots, n; \ \forall k = 1, \dots, m) \qquad (3m)$$

$$W_j - W_{(0,k)} \leq -d_j^{(w)} + (Sc_j^{ini} + d_j^{(c)} - Sc_j^{end}) * \beta + (1 - x_{0j}^{k}) M_{13}$$
$$(\forall j = 1, \dots, n; \ \forall k = 1, \dots, m) \qquad (3n)$$

$$C_j - C_{(i)} \geq (Sc_j^{ini} - Sc_j^{end}) + (\sum_{k=1}^{k} x_{0j}^{k} - 1) M_{14}$$
$$(\forall i, j = 1, \dots, n; i \neq j) \qquad (3o)$$

$$C_j - C_{(i)} \leq (Sc_j^{ini} - Sc_j^{end}) + (1 - \sum_{k=1}^{k} x_{0j}^{k}) M_{14}$$
$$(\forall i, j = 1, \dots, n; i \neq j) \qquad (3p)$$

$$C_{(0,k)} = \sum_{j=1}^{n} d_j^{(c)} \sum_{i=0_{i \neq j}}^{n} x_{ij}^{k} \quad (\forall k = 1, \dots, m) \qquad (3q)$$

$$C_j - C_{(0,k)} \geq (Sc_j^{ini} - Sc_j^{end}) + (x_{0j}^k - 1)M_{17}$$

$$(\forall j = 1, ..., n; \ \forall k = 1, ..., m) \qquad (3r)$$

$$C_j - C_{(0,k)} \leq (Sc_j^{ini} - Sc_j^{end}) + (1 - x_{0j}^k)M_{17}$$

$$(\forall j = 1, ..., n; \ \forall k = 1, ..., m) \qquad (3s)$$

$$Sc_j^{end} \leq Sc_j^{ini} + d_j^{(c)} \quad (\forall j = 1, ..., n) \qquad (3t)$$

$$U_j - U_i + \sum_{k=1}^{m} \left[ (M_{20} + d_i^{(w)})x_{ij}^k + (-d_i^{(w)} + M_{20})x_{ij}^k \right] \leq M_{20}$$

$$(\forall i, j = 0, ..n; i \neq j) \qquad (3u)$$

$$d_j^{(w)} \leq U_j \leq \sum_{k=1}^{m} \sum_{i=0_{i\neq j}}^{n} Q_k^{(w)} x_{ij}^k (\forall j = 1, ..n) \qquad (3v)$$

Knowing that:

- $TS_j = TS_{(u)} * (2d_j^{(c)} + Sc_j^{ini} - Sc_j^{end}) + TS_{(f)}$
- $TC_j = TPc_{(u)} * d_j^{(c)}$
- $M_9 = M_{10} = M_{12} = M_{13} = M_{20} = Max_{k'}Q_{k'}^{(w)}$
- $M_{14} = M_{15} = M_{17} = M_{18} = Max_{k'}Q_{k'}^{(c)}$

It is worth noting that *MILP*3 and *MILP*1 are quite similar. As mentioned earlier in this section, *MILP*3 allows leaving some boxes due to either vehicle capacity or trip cost. To satisfy this flexibility, we added constraint (3s), which ensures that boxes related to customer demand plus initial balance must not exceed total required boxes.

## 5. Computational Study

We adopted a configuration quite similar to the one considered in [29], where authors have implemented their models in the IBM ILOG CPLEX Optimization Studio (Version: 12.6). All the experiments were conducted on a computer with an Intel(R) Core (TM) i7-7700 CPU@3.6 GHz and 8 GB memory under the Windows 10 Pro system. As we highlighted earlier in the introduction section, our work is dedicated to a poultry company in Tunisia, called CHAHIA. Due to a confidentiality issue, CHAHIA has fed our research work with limited data. Based on the collected data, we created a set of instances. To make sure that data features are diversified, we created 44 instances. Several factors have been used to calibrate the quality of solutions, enhance the solving process, and diversify the benchmark of instances. These factors are as follows:

- The types of demands, i.e., the quantities of required goods (low/medium/high).
- The number of boxes for customers (delivery and initial balance) and their dispersion.
- The number of customers to be served.
- The geographic dispersion of customer locations (distance between customers).
- The number of vehicles.
- The vehicle capacities: we considered, small (S) with 1.4 tons, and big (B) with 3 tons.

Tables 1–3 represent the computational results provided by our proposed models. The results are shown in terms of objective function value (cost), computational time to reach solutions, and number of iterations.

Results show that *MILP*2 achieves the lowest objective function values compared to both *MILP*1 and *MILP*3. This achievement is mainly due to its flexibility and high number of specifications that it covers.

In case of low demand or low number of boxes to be picked up, both *MILP*1 and *MILP*3 provide similar costs. However, if the demand or box numbers increase, *MILP*3 outperforms *MILP*1 and provides the lowest travel cost. This fact shows that sometimes it would be more profitable to leave some boxes instead of collecting them. We recall that the left boxes could be collected in future trips, otherwise, a penalization is considered.

In case we reduce the cost of left boxes by relying on the possibility to recover them in the future with better conditions, *MILP*3 will provide the lowest cost for the majority of instances. Moreover, if we change the capacity of vehicles from small to big, such as in instances "inst5.2.004" and "inst5.2.005", which are the same in fact, *MILP*1 and *MILP*3 diversify and provide different value costs, however, *MILP*2 remains unchangeable. Furthermore, if we increase both, the demand to be between 151 Kg and 500 Kg per customer, and the number of customers to exceed 18, *MILP*3 becomes the most efficient model. With regard to the computational time performance indicator, *MILP*3 converges quickly. Contrariwise, *MILP*2 consumes more time to provide optimal solutions. In case the number of customers exceeds six, *MILP*2 remains incapable of retrieving an optimum solution during a reasonable period of time (i.e., less than 3600 min). With regard to the number of served customers, *MILP*1 and *MILP*3 provide a better performance compared to *MILP*2.

It is noticeable that when the number of vehicle stops for pickup or delivery increase, it will affect the quality of poultry products from CHAHIA. This affection is related to the unstable air conditioning of the vehicle refrigeration cabin. Therefore, it is recommended to reduce the number of load breaks.

The different readings of the same problem led to an interesting diversification in terms of travel cost, computational time, and number of served customers.

**Table 1.** Low demand and customers number instances.

| n | m | Type | Instances | MILP1 | | | MILP2 | | | MILP3 | | |
|---|---|------|-----------|-------|---|---|-------|---|---|-------|---|---|
| | | | | OF (TND) | CPU (Sec) | Nodes | OF (TND) | CPU (Sec) | Nodes | OF (TND) | CPU (Sec) | Nodes |
| 5 | 1 | S | Inst5.1.001 | 1.780 | 0.03 | 49 | 1.200 | 0.48 | 589 | 1.780 | 0.03 | 52 |
| | - | B | Inst5.1.002 | 1.780 | 0.08 | 55 | 1.200 | 4.32 | 5701 | 1.780 | 0.09 | 55 |
| | 2 | S | Inst5.1.003 | 1.780 | 0.1 | 128 | 1.200 | 39.17 | 16,901 | 1.780 | 0.1 | 133 |
| | - | B | Inst5.1.004 | 2.660 | 0.3 | 528 | 1.200 | 53.83 | 26,082 | 2.660 | 0.3 | 687 |
| | - | B | Inst5.1.005 | 1.780 | 0.1 | 127 | 1.200 | 67.72 | 19,848 | 1.780 | 0.1 | 145 |
| 5 | 1 | S | Inst5.1.006 | 3.980 | 0.09 | 38 | 2.770 | 2729.4 | 5,099,447 | 3.980 | 0.09 | 37 |
| | - | B | Inst5.1.007 | 4.030 | 0.06 | 47 | 2.770 | 251.3 | 427,744 | 3.980 | 0.08 | 24 |
| | 2 | S | Inst5.1.008 | 3.980 | 0.05 | 32 | 2.770 | 309.8 | 410,657 | 3.980 | 0.09 | 32 |
| | - | B | Inst5.1.009 | 4.700 | 0.2 | 260 | 2.770 | 3899.9 | 2,764,520 | 3.980 | 0.17 | 75 |
| 6 | 1 | S | Inst6.1.001 | 2.200 | 0.1 | 155 | 1.510 | 267 | 232,811 | 2.200 | 0.2 | 173 |
| | - | B | Inst6.1.002 | 2.200 | 0.1 | 139 | 1.510 | 0.8 | 405 | 2.200 | 0.1 | 158 |
| | 2 | S | Inst6.2.003 | 2.600 | 1 | 1630 | 1.510 | 42.9 | 13,972 | 2.200 | 0.5 | 668 |
| | - | B | Inst6.2.004 | 3.010 | 0.9 | 1263 | 1.510 | 499.4 | 230,267 | 2.600 | 1 | 1515 |
| | - | B | Inst6.2.005 | 3.010 | 1.3 | 2401 | 1.510 | 255.6 | 101,203 | 2.200 | 0.3 | 366 |

**Table 2.** Medium demand and moderate number of customers instances.

| n | m | Type | Instances | MILP1 | | | MILP2 | | | MILP3 | | |
|---|---|------|-----------|-------|---|---|-------|---|---|-------|---|---|
| | | | | OF (TND) | CPU (Sec) | Nodes | OF (TND) | CPU (Sec) | Nodes | OF (TND) | CPU (Sec) | Nodes |
| 8 | 1 | S | Inst8.1.001 | 3.810 | 9.4 | 13,954 | - | - | - | 3.818 | 9.2 | 13,726 |
| | - | B | Inst8.1.002 | 3.810 | 1.3 | 1275 | - | - | - | 3.818 | 7 | 12.425 |
| | 2 | S | Inst8.2.003 | 5.770 | 81.6 | 92,173 | - | - | - | 5.580 | 145.7 | 187,467 |
| | - | B | Inst8.2.004 | 10.750 | 120.7 | 137,284 | - | - | - | 7.540 | 515.7 | 506,652 |
| | 3 | S | Inst8.3.005 | 5.580 | 220.8 | 299,822 | - | - | - | 3.810 | 47.5 | 60,144 |
| | - | B | Inst8.3.006 | 5.580 | 138.5 | 12,515.5 | - | - | - | 5.580 | 349 | 410,268 |
| 10 | 2 | S | Inst10.2.001 | 2.870 | 17 | 1810 | - | - | - | 2.750 | 25.32 | 26,275 |
| | - | B | Inst10.2.002 | 3.170 | 394.5 | 418,597 | - | - | - | 3.160 | 2030.4 | 2,181,194 |
| | 3 | S | Inst10.3.003 | 3.170 | 6176.4 | 3,682,568 | - | - | - | 2.750 | 242.4 | 196,425 |
| | - | B | Inst10.3.004 | 3.170 | 1619.2 | 1,134,364 | - | - | - | 3.160 | 5337.8 | 3,738,052 |
| 12 | 2 | S | Inst12.2.001 | 3.270 | 3464 | 2,127,116 | - | - | - | 2.860 | 57.2 | 32,994 |
| | - | B | Inst12.2.002 | 4.020 | 9002.5 | 7,765,020 | - | - | - | 2.860 | 45 | 37,430 |
| | 3 | S | Inst12.3.003 | 3.270 | 5327.2 | 2,369,060 | - | - | - | 2.860 | 244.9 | 92,847 |
| | - | B | Inst12.3.004 | 3.730 | 39,770.9 | 24,454,247 | - | - | - | 2.860 | 101.25 | 51,634 |

**Table 3.** High demand and customers number instances.

| n | m | Type | Instances | MILP1 | | | MILP2 | | | MILP3 | | |
|---|---|---|---|---|---|---|---|---|---|---|---|---|
| | | | | OF (TND) | CPU (Sec) | Nodes | OF (TND) | CPU (Sec) | Nodes | OF (TND) | CPU (Sec) | Nodes |
| 15 | 2 | S | Inst5.2.001 | 5.380 | 24,453.7 | 12,171,800 | - | - | - | 5.060 | 1110.6 | 654,979 |
| | - | B | Inst5.2.001 | 5.5510 | 29,199.2 | 16,852,192 | - | - | - | 5.060 | 437 | 2,448,044 |
| | 3 | S | Inst5.3.003 | 5.060 | 2714.4 | 1,311,100 | - | - | - | 5.060 | 2548.8 | 835,035 |
| 18 | 2 | S | Inst18.2.001 | 3.630 | 4959.2 | 1,834,815 | - | - | - | - | - | - |
| | - | B | Inst18.2.002 | - | - | - | - | - | - | 3.630 | 4475.7 | 1,830,280 |
| | 3 | S | Inst18.3.003 | 3.630 | 9512.2 | 3,381,239 | - | - | - | - | - | - |
| | - | B | Inst18.3.004 | - | - | - | - | - | - | 3.630 | 6081.3 | 1,780,109 |
| 20 | 2 | S | Inst20.2.001 | 3.580 | 916.1 | 254,457 | - | - | - | - | - | - |
| | - | B | Inst20.2.002 | - | - | - | - | - | - | - | - | - |
| | 3 | S | Inst20.3.003 | 3.580 | 4472.9 | 893,970 | - | - | - | 3.580 | 4208.4 | 1,128,435 |
| | - | B | Inst20.3.004 | - | - | - | - | - | - | - | - | - |
| 23 | 2 | S | Inst23.2.001 | 4.290 | 4396.2 | 851,800 | - | - | - | - | - | - |
| | - | B | Inst23.2.002 | - | - | - | - | - | - | 4.290 | 26,413.7 | 3,834,209 |
| | 3 | S | Inst23.3.003 | - | - | - | - | - | - | - | - | - |
| | - | B | Inst23.3.004 | - | - | - | - | - | - | - | - | - |

## 6. Discussion of Limitations

Our contribution develops a *'proof of concept'* useful to establish better awareness and foresight in decision-making processes. It provides flexibility in reacting against miscellaneous realistic situations in transportation and goods delivery activities. However, the major limitations of this work could be summarized as follows:

- Due to a confidentiality issue, CHAHIA has fed our research work with limited data. Therefore, bigger data is required for a more relevant benchmark.
- CHAHIA has refused to reveal all serve stores, customers, and quantities, due to confidentiality.
- CHAHIA was looking for a prototype that could be enhanced in the future by its engineers.
- Due to lack of equipment, we were not able to perform any experiments with more than 23 customers for *MILP*1 and *MILP*3, and 6 customers for *MILP*2.
- Exact methods considered in this work are incapable of solving large instances. Therefore, integrating heuristics is highly recommended.

## 7. Conclusions and Future Work

In this paper, the Vehicle Routing Problem (VRP) with Pickup and Delivery (VRPPD) was addressed based on real-life perceptions. Three different VRPPD models have been developed based on Mixed Integer Linear Programming approaches. The first model is designed as a VRP with simultaneous pickup and delivery. The second one is modeled as a VRP with either simultaneous and/or separated pickup and delivery. The last one is modeled as a VRP with simultaneous and flexible delivery and pickup allowing both partial pickups and penalized losses. In addition, this study proposes a *proof of concept* based on a rich benchmark. The data adopted in this work were provided by a poultry company in Tunisia, called CHAHIA. The developed models have been compared against each other using multiple state-of-the-art key performance indicators. Based on the presented richness in term of modeling, interesting perspectives could be considered in the future, such as matching operational issues with tactical planning elements in order to make the models more dynamic and robust facing various demand scenarios. Finally, various heuristics could be proposed in order to solve large instances and enable quicker convergence towards the optimum.

**Author Contributions:** Conceptualization, A.L. and R.M.; methodology, A.L.; validation, R.L. and A.A.; experimentation (coding and data collection): A.L.; writing—review and editing, A.L.; project administration, A.L. and M.N.; funding acquisition, A.L. All authors have read and agreed to the published version of the manuscript.

**Funding:** Deanship of Scientific Research at Prince Sattam bin Abdulaziz University. Research project No. 2020/01/13222.

**Institutional Review Board Statement:** Not applicable.

**Informed Consent Statement:** Not applicable.

**Data Availability Statement:** Data is available on demand from the corresponding author.

**Acknowledgments:** This project was supported by the Deanship of Scientific Research at Prince Sattam bin Abdulaziz University through the research project No. 2020/01/13222.

**Conflicts of Interest:** The authors declare no conflict of interest.

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
