# Peer review of "Mixed Integer Linear Programming Models to Solve a Real-Life Vehicle Routing Problem with Pickup and Delivery"

_applsci, doi:10.3390/app11209551_

Round 1
Reviewer 1 Report
There are countless papers on mathematical formulations using MIP models for Pickup and Delivery Vehicle Routing Problems. More complex variants including time windows, backhauls, etc. have been already handled using this framework. The authors should clearly establish the contributions of their study. Moreover, the authors should use benchmark instances of Pickup and Delivery problems to strengthen their case (instead of relying on smaller custom instances). The paper also needs to be proofread in order to minimize grammatical errors and stylistic issues.
Moreover, the authors present three different MIP models which are not novel based on different variable definitions. If the problem contained real life constraints, then it would be more meaningful. In the current form, the draft considers a very well-studied Pickup and Delivery problem and presents three different MIP models for it. Note that this Pickup and Delivery problem is well documented with years of research. Furthermore, the authors perform preliminary experiments on very small datasets without any regards to differences and utility of three proposed models. The paper could be improved by experiments on benchmark instances and comparison with standard heuristics. Their whole computational experiments section is around 2 pages. In summary, the paper's contributions are unclear and the paper is also full of grammatical errors and stylistic issues.
Author Response
We would like to thank the reviewer for the time and effort s/he spent evaluating our article. In the attached document, detailed answers and rebuttals to these comments are provided. Moreover, we have updated the manuscript to accommodate these comments.

Reviewer 2 Report
- The paper must be fully developed - includes discussion, contribution, implication and limitations.
- I would like to see a well-developed discussion (minimum two pages) comparing and contrasting solution/results presented in the work with existing work and then a subsection of it presenting contributions to theory/knowledge/literature (at least one to two paragraphs) and followed by a subsection on Implications for practice (at least one page). In these paragraphs authors should compare their research approach with previous research, citing references of others' research.
- The conclusion section must have a subsection on limitations and future research directions (one to two pages).
- The overall document should be checked for grammar, syntax, and typos errors.
- Authors should reconsider to explain the section about the scientific contribution in the introduction, as well as in the conclusion part of the paper, with the structured comparison of the current research with previous research. The text can be one paragraph long, but it should contain the most important studies.
- Please, form the conclusion in the following manner: (i) First paragraph - summary of research and conclusion - e.g. In this paper... ; (ii) Second paragraph - comparison with previous research; (iii) Third paragraph - short description of practical implications; (iv) Fourth paragraph - summary of paper limitations and further implications.
Author Response

(The authors gave the same response as above.)

Reviewer 3 Report
I have following concerns for the authors to address.
1. In the literature review, after introducing existing articles on VRP, the authors need to indicate the research gaps between the current research and the topic. Then, the authors need to show their works and how they can bridge the gaps. Accordingly, the authors need to summarize their contributions.
2. The articles reviewed should be updated. The reviewed articles are out of date.
3. The authors need to give the reasons why three models are formulated. What are their interrelations in dealing with the VRP.
4. The authors need to show the formulation of their models (integer or mixed integer, linear or nonlinear) after explaining the representations of the objectives and constraints. This are closely related to the feasibility of using mathematical programming software to solve the problem, which is stressed by many studies, e.g., Sun (2020, https://doi.org/10.1007/s40815-020-00905-x) and Gao and Cao (2020, https://doi.org/10.1016/j.cie.2020.106782). Referring to above studies, the authors need to associate the use of the mathematical programming software with the formulations of their models at the beginning of Section 5.
5. The authors did not give the experimental case in detail. As indicated by the table at the end of Section (please add its title), there are cases where using mathematical programming software is time consuming. Therefore, considering the NP hardness of the VRP, the authors need to consider to use heuristic algorithms.
6. The results findings from the experiment need to be listed one by one at the end of Section 5.
7. This paper is oriented on an agri-food sector. However, the VPR modeling in this paper is too general and doesn’t illustrate the unique characteristics of deliveries and pickups under this background.
8. The conclusions are simple. The authors need to focus on their unique works and contributions. The limitations should also be explained and corresponding future works should then be discussed.
Author Response

(The authors gave the same response as above.)

Round 2
Reviewer 1 Report
Please carefully proofread the paper. Here are some suggestions for improving readability of the paper:
- Use "computational experiments" instead of "an intensive computational simulations" in the Abstract.
- Use "routing plans" instead of "routing pacification" (P.1)
- Rephrase "boxes are becoming more consumable" (P.2)
- Rephrase last sentence preceding See Figure 1 (P.2)
- Rephrase "problem apprehension" in Figure 1
- vistes - P.3
- Page 4 - Incorporate spaces in the bullet points like Time Windows
- Page 4 - Use "addressed" instead of "evoked"
- Page 15 - 6 costumes
- Remove "using multiple state-of-the-art key performance indicators". Instead use "using smaller custom datasets" in Conclusions
- Use "interesting perspectives could be considered in the future"
- Use "facing various demand scenarios" instead of "various a demand"
- Use "Finally, various heuristics could be proposed in order to solve …"
Author Response
We thank the reviewer for his assistance to enhance the paper quality and fix some mistakes. We provided a detailed letter addressing his comments.

Reviewer 3 Report
The authors addressed my comments in a good manner.
Author Response
We thank reviewer #3 for acknowledging our updates and answers associated with his comments.
This manuscript is a resubmission of an earlier submission. The following is a list of the peer review reports and author responses from that submission.